# Mendelian randomization as a tool to inform drug development using human genetics

Iyas Daghlas[1] and Dipender Gill[2,3] 

[1]Department of Neurology, University of California San Francisco, San Francisco, CA, USA; [2]Department of Epidemiology and Biostatistics, School of Public Health, Imperial College London, London, UK and [3]Chief Scientific Advisor Office, Research and Early Development, Novo Nordisk, Copenhagen, Denmark

drug development; drug targets; GWAS; IL-6; Mendelian randomization

**Author for correspondence:**
Iyas Daghlas,
Email: iyas.daghlas@ucsf.edu

## Abstract

Drug development is essential to the advancement of human health, however, the process is slow, costly, and at high risk of failure at all stages. A promising strategy for expediting and improving the probability of success in the drug development process is the use of naturally randomized human genetic variation for drug target identification and validation. These data can be harnessed using the Mendelian randomization (MR) analytic paradigm to proxy the lifelong consequences of genetic perturbations of drug targets. In this review, we discuss the myriad applications of the MR paradigm for human drug target identification and validation. We review the methodology and applications of MR, key limitations of MR, and potential future opportunities for research. Throughout the review, we refer to illustrative examples of MR analyses investigating the consequences of genetic inhibition of interleukin 6 signaling which, in some cases, have anticipated results from randomized controlled trials. As human genetic data become more widely available, we predict that MR will serve as a key pillar of support for drug development efforts.

## Impact statement

Mendelian randomization (MR) is a method that uses naturally randomized human genetic variation to study the lifelong effects of genetic perturbations of drug targets. This approach has great promise to help speed up and improvef the drug development process. In this review, we discuss how MR is used for identifying and testing drug targets, its limitations, and future opportunities for research. As more human genetic data become available, we expect MR to play a major role in drug development.

## Introduction

The last century has seen major advances in pharmacotherapy within all medical specialties and with consequent reductions in morbidity and mortality (Fuchs, 2010; Lichtenberg, 2019). Despite this, there remain many unmet medical needs that necessitate ongoing drug development efforts. The challenges inherent to the process of drug development are highlighted by the poor success rate of drug development programs, which has been estimated to be as low as four percent (Hay et al., 2014; Finan et al., 2017). Challenges contributing to this high failure rate include substantial costs (DiMasi et al., 2003; Schlander et al., 2021), a low probability of passing preclinical testing (van Norman, 2019), poor concordance between efficacy in preclinical studies and clinical trials (Perel et al., 2007), and limitations in accurately predicting drug toxicity in human (Bailey et al., 2014; Paglialunga et al., 2019; van Norman, 2019). A common theme underlying these challenges is the poor translatability of findings from animal models to humans (Akhtar, 2015).

These limitations of preclinical data in predicting drug efficacy and toxicity have motivated the use of alternative strategies for drug target selection and validation. An increasingly popular strategy is to leverage human genetic variation influencing protein-coding genes, as most small molecules and biologics target proteins (Santos et al., 2016). There are several features of human genetic variation that make it an attractive source of data for this endeavor (Figure 1). First, germline genetic variants are randomly allocated at gametogenesis (Davies et al., 2018). Thus, if confounding by genetic ancestry is appropriately controlled for (Price et al., 2006), the inheritance of genetic variants is not confounded by environmental variables. This is in contrast to conventional observational analyses that investigate efficacy or repurposing potential for drugs used in clinical practice (also referred to as pharmacoepidemiologic analyses). Second, the assignment of a germline genetic variant is fixed at gametogenesis and is therefore not susceptible to change following development of disease. This phenomenon of reverse causality commonly biases pharmacoepidemiologic analyses. These two inherent properties allow for genetic variants to be leveraged in naturally randomized experiments that can increase confidence in the causal

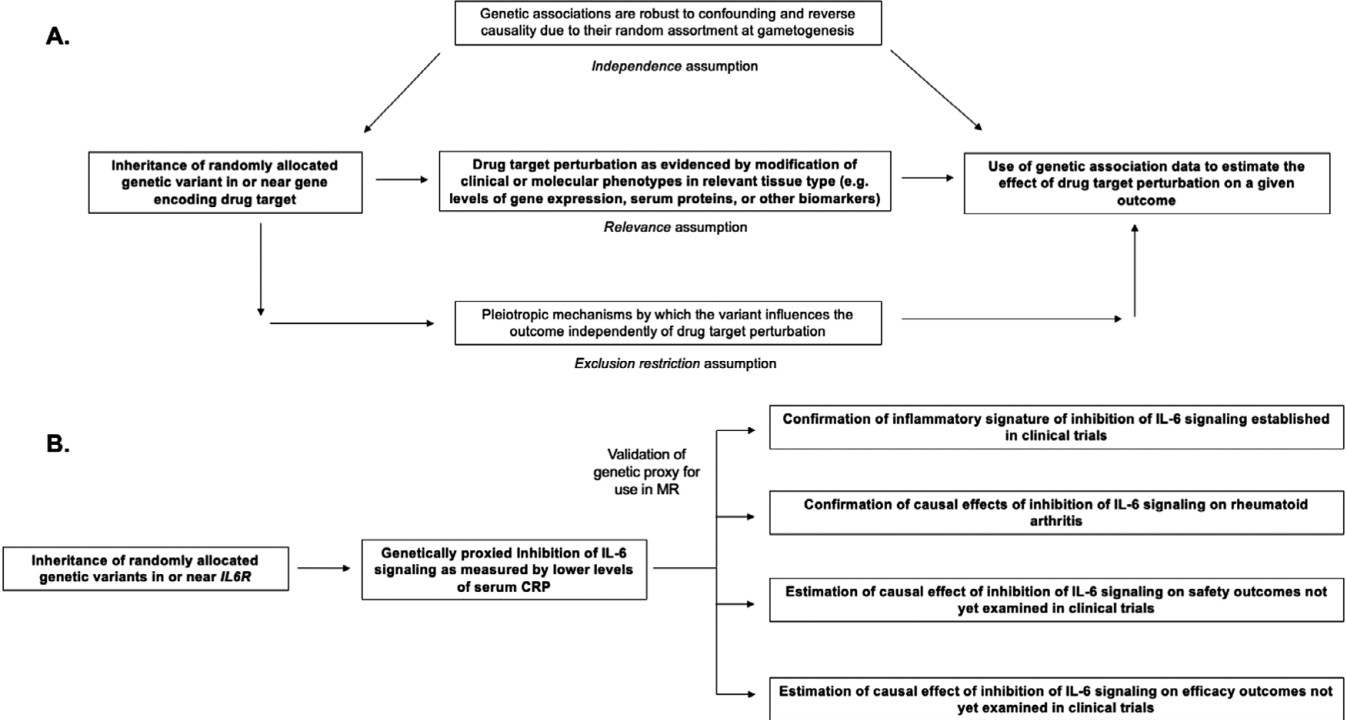

**Figure 1.** Structure and assumptions of Mendelian randomization analyses (A) and application to the example of genetically proxied inhibition of interleukin 6 (IL-6) signaling (B).

relationship between an exposure and an outcome of interest (Davies et al., 2018). Third, human genetic variants can be linked to clinically relevant efficacy and safety outcomes in humans rather than in model organisms. Fourth, large genetic datasets from millions of study participants are readily available for a wide variety of clinically relevant phenotypes including clinical risk factors, disease outcomes, circulating proteins, and diverse imaging phenotypes (Sudlow et al., 2015; Elliott et al., 2018; Hemani et al., 2018; Kurki et al., 2022). In many cases, these data are publicly available, and statistical methods may be used to combine information from different datasets, thus rendering their use efficient and highly cost-effective.

The potential for human genetics to inform drug development is exemplified by the development of PCSK9 inhibitors, a cholesterol-lowering drug approved in record time for the prevention of coronary artery disease (CAD) (Hall, 2013). This drug target was first discovered on the basis of damaging variants in the *PCSK9* gene that reduced circulating low-density lipoprotein (LDL) cholesterol levels and reduced the risk of CAD (Cohen et al., 2006). In effect, these variants served as proxies for the potential clinical benefits of LDL-cholesterol lowering via PCSK9 inhibition. The safety profile of PCSK9 inhibition was corroborated by identification of a patient with complete genetic inactivation of *PCSK9* due to loss-of-function variants (Hall, 2013). Careful phenotyping of this patient identified no major health consequences and served as proof-of-concept for the safety of complete PCSK9 inhibition. PCSK9 inhibitors, such as evolocumab and alirocumab, were subsequently approved (Shapiro et al., 2018), and have demonstrated consistent efficacy in clinical trials for the lowering of LDL-cholesterol and prevention of CAD (Karatasakis et al., 2017; Shapiro et al., 2018).

Although the success of human genetics in identifying PCSK9 as a promising drug target is an outlier, there is broader evidence that

drug targets with genetic support are more likely to be approved (Nelson et al., 2015; King et al., 2019). This was initially evidenced by a seminal analysis (Nelson et al., 2015) that matched drug-disease pairs with genetic association-phenotype pairs and determined that approved drugs were more likely to have supporting genetic mechanisms. Moreover, drug targets with this form of genetic support had higher odds of progression within each phase of a clinical trial (e.g., phase 1 to 2, phase 2 to 3). It was estimated that selecting a target with genetic support could double the success rate for drugs in clinical development. This finding was replicated in a subsequent study which also found that the strongest form of genetic evidence was in the form of genetic variation that impacts the protein-coding sequence of a gene (King et al., 2019). Of note, these retrospective analyses did not select for drug target programs that were initially motivated or supported by genetic data prior to approval of the drug. Thus, although these data are encouraging, it remains unclear the extent to which human genetic support may prospectively increase the odds of a successful drug program. Moreover, these estimates were obtained prior to the widespread use of the methods detailed in this review and may therefore be an underestimate of the true utility of genetics for drug discovery.

## Mendelian randomization

Mendelian randomization (MR) is the paradigm by which randomly assorted germline genetic variants can be used as proxies for a disease risk factor or for drug target perturbation (Smith and Ebrahim, 2003, 2004; Davies et al., 2018; Burgess et al., 2019). Specifically, MR leverages naturally randomized genetic variants as proxies (also referred to as *instrumental variables*) for modifying a given exposure to test causal effects on an outcome of interest. The features of human genetic variation outlined above – namely the

random assortment at gametogenesis that renders genetic variants less susceptible to confounding and to reverse causality – increase confidence in causal inference from MR analyses (Davies et al., 2018). Several key assumptions are necessary for causal inference in this paradigm, including that the genetic variants used as proxies are strongly associated with the exposure of interest (the *relevance* assumption), that the association of the genetic variants with the exposure and with the outcome are not confounded by environmental variables or by nearby genetic signals (the *independence* assumption), and that the association of the genetic variants with the outcome is not explained by pathways independent of the exposure of interest (also known as pleiotropy) (Davies et al., 2018). These assumptions are summarized in Figure 1. A discussion of the impact of confounding due to nearby genetic signals and the statistical method of colocalization for overcoming this bias is discussed at length by Zuber et al. (2022). The last assumption, also referred to as the *exclusion restriction* condition, will be discussed later in this review.

An MR analysis may be conducted when a genetic variant(s) has met the above criteria to be used to proxy a drug target. In the simplest form of an MR analysis, these genetic proxies may be directly tested for their association with a clinical outcome of interest (Gill et al., 2021). For *PCSK9*, this involved testing for an association of the genetic variants with risk of CAD. Statistical methods may be implemented to weigh the effect on the outcome by a unit increase in the biomarker of interest, such as mmol/L of LDL-cholesterol (Gill et al., 2021).

The *PCSK9* example exemplifies these key principles for the application of MR for drug development. First, the genetic variants proxying PCSK9 inhibition were strongly associated with lower LDL-cholesterol (Cohen et al., 2006). These variants, therefore, satisfy the MR assumption of *relevance.* Second, the variants were positioned within the protein-coding sequence of *PCSK9* and could thus more confidently, but not definitively, be predicted to exert their effects via influencing *PCSK9* function. This contrasts with intronic variants in or around the gene that are more likely to tag causal variants in nearby genes (Acosta et al., 2021). Third, the genetic variants in *PCSK9* were not associated with confounding

variables such as age, sex, smoking status, and type 2 diabetes. This supports the MR independence assumption of no confounding and the exclusion restriction assumption that the variants do not exert pleiotropic effects via these pathways.

## Scope of review

The recent years have seen an explosion in the methodology and application of drug target-focused MR (Acosta et al., 2021; Gill et al., 2021). In this review, we discuss applications and case studies of these methodologies and approaches. The sections are structured to highlight how MR may be used to provide guidance and supporting data at all stages of drug development (Figure 2). It is our hope that this review will serve as an accessible resource for understanding applications and key limitations of MR for informing drug development.

Throughout, we incorporate examples from genetic analyses investigating inhibition of interleukin 6 (IL-6) signaling, a pleiotropic inflammatory cytokine that is the target of Food and Drug Administration-approved drugs including tocilizumab, sarilumab, and satralizumab (Figure 1; Kang et al., 2019). These genetic analyses (summarized in Table 1) serve as examples for how MR may inform all stages of drug development, including anticipation of results from ongoing clinical trials investigating inhibition of IL-6 signaling for the prevention of cardiovascular disease and preservation of kidney function (Ridker and Rane, 2021). We focus on IL-6 as we anticipate that, in contrast to the established example of PCSK9 inhibitors, these ongoing trials will serve as prospective tests for how MR may inform drug development. Relevant citations were identified in the MEDLINE database using the keyword combinations "'IL-6" + "Mendelian randomization"' and "'interleukin 6" + "Mendelian randomization"'. The present study was not designed as a systemic review.

## Planning the analysis: Drug target discovery or selection

A drug target may first be identified from a genome (GWAS) or exome-wide association study conducted using a clinical phenotype

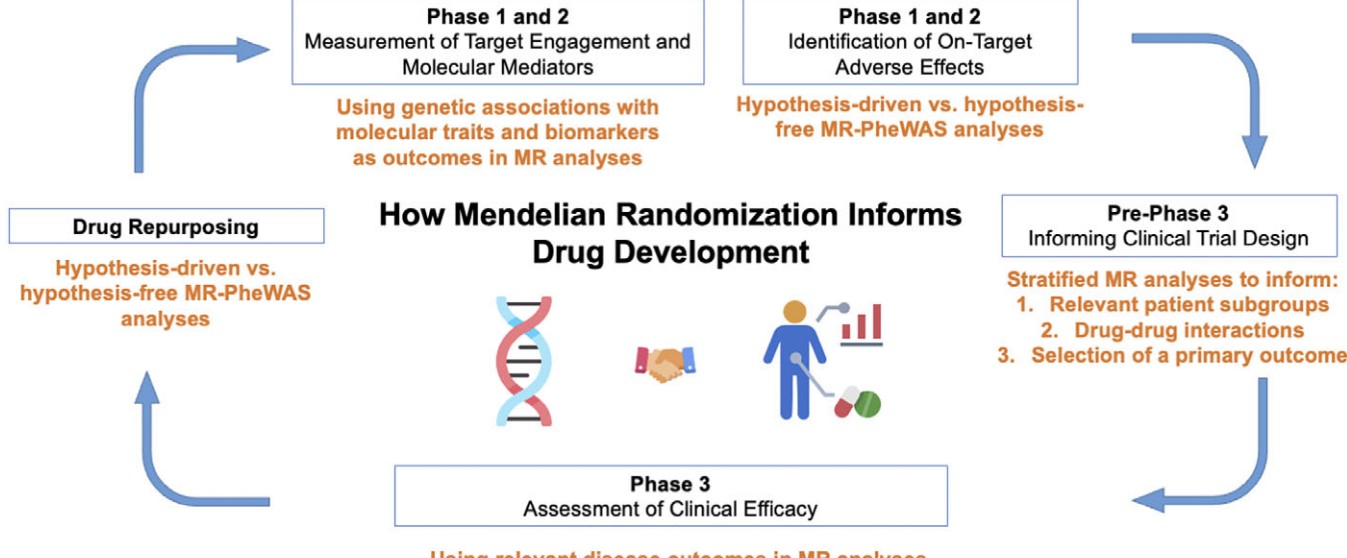

**Figure 2.** Applications of Mendelian randomization (MR) to each phase of drug development. PheWAS, phenome-wide association study.

**Table 1.** Selected examples of MR analyses cited in this Review that investigated effects of genetically proxied inhibition of IL-6 signaling on a diverse range of outcomes

| Study citation (year) | Genetic proxy for inhibition of IL-6 signaling | Primary outcome(s) | Key secondary outcomes | Effect of genetically proxied inhibition of IL-6 signaling | Parallel clinical trial stages |
|---|---|---|---|---|---|
| Rahman et al. (2022)[62] | 30 genetic variants *cis* to the *IL6R* gene and influencing levels of CRP | 40 circulating cytokines | NA | Reduced levels of 10 cytokines | Phase 1/2 – Target engagement and identifying molecular signatures |
| Larsson et al. (2021)[83] | 7 genetic variants *cis* to the *IL6R* gene and influencing levels of CRP | COVID-19 susceptibility, severity, and hospitalization | Pneumonia | Protective for COVID-19 susceptibility, severity, and hospitalization Increases risk of pneumonia | Phase 1/2 – Toxicity Phase 3 – Efficacy Repurposing |
| Rosa et al. (2019)[60] | 34 genetic variants associated with soluble IL-6 receptor (a negative regulator of IL-6 signaling) | 8 CVD outcomes | Rheumatoid arthritis, atopic dermatitis, asthma, longevity | Protective for all cardiovascular outcomes and for rheumatoid arthritis. Associated with longer lifespan. Adverse effects on atopic dermatitis and asthma. | Phase 1/2 - Toxicity Phase 3 – Efficacy Repurposing |
| Georgakis et al. (2021)[54] | 7 genetic variants *cis* to the *IL6R* gene and influencing levels of CRP | PheWAS | | Protective for CAD, AAA, T2D, and varicose veins Increased risk for infection and atopic dermatitis | Phase 1/2 – Toxicity Repurposing |
| Georgakis et al. (2022)[69] | 26 genetic variants *cis* to the *IL6R* gene and influencing levels of CRP | Combined CVD endpoint: CAD, ischemic stroke, AAA, and PAD within population subgroups | NA | Greater relative benefit in individuals with high LDL-cholesterol. Greatest absolute benefit in individuals with high baseline CRP levels | Phase 3: informing clinical trial design in relation to population subgroups |
| Georgakis et al. (2022)[72] | 26 genetic variants *cis* to the *IL6R* gene and influencing levels of CRP. This genetic proxy was stratified across levels of genetically proxied *HMGCR* inhibition to query potential interactions of IL-6 inhibitors with statin therapy. | Combined CVD outcome: CAD, ischemic stroke, PAD, AA, cardiovascular death | | Genetically proxied inhibition of IL-6 signaling and of *HMGCR* additively reduce CVD risk | Phase 3: informing clinical trial design in relation to drug-drug interactions |
| Sarwar et al. (2012)[38] | A single genetic variant: rs2228145 - Asp358Ala (reduces membrane bound IL-6 receptor and thus increases serum IL-6 receptor levels) | CAD | NA | Protective for CAD | Phase 3 - Efficacy Repurposing |
| Georgakis et al. (2020)[59] | 7 genetic variants *cis* to the *IL6R* gene and influencing levels of CRP | Ischemic stroke, CAD | AAA, AF, heart failure, thromboembolism, PAD, carotid plaque | Protective for CAD, ischemic stroke, AAA, AF, and carotid plaque. Null effects for heart failure and thromboembolism, and PAD. | Phase 3 – Efficacy Repurposing |
| Toshner et al. (2022)[76] | rs7529229, a variant that tags the variant used by Sarwar et al. above | Pulmonary arterial hypertension (PAH) | Mortality in PAH | Null effect on PAH | Phase 3 – Efficacy Repurposing |
| Zhao and Gill (2022)[80] | 7 genetic variants *cis* to the *IL6R* gene and influencing levels of CRP | Polymyalgia rheumatica (PMR) | NA | Protective for PMR | Phase 3 – Efficacy Repurposing |
| Bovjin et al. (2021)[84] | 7 genetic variants *cis* to the *IL6R* gene and influencing levels of CRP | COVID-19 susceptibility, severity, hospitalization | NA | Protective for COVID-19 susceptibility, hospitalization, but not severity | Phase 3 – Efficacy Repurposing |
| Cai et al. (2018)[85] | rs2228145, as described above | PheWAS | | Protective for CAD and AAA | Repurposing |
| Levin et al. (2021)[75] | rs2228145 | PAD and disease subtypes | NA | Protective for PAD and all subtypes | Phase 3 – Efficacy Repurposing |

*Abbreviation*: AA: aortic aneurysm; AAA: abdominal aortic aneurysm; AF: atrial fibrillation: CAD: coronary artery disease; CRP: c-reactive protein; CVD: cardiovascular disease; PAD: peripheral artery disease; PheWAS: phenome-wide association study; T2D: type 2 diabetes.

of interest. For example, a 2013 GWAS of CAD identified a lead variant in the *IL6R* gene (Deloukas et al., 2013). This finding, in the context of independent human genetic evidence for a role of IL-6 signaling in the development of cardiovascular disease (Sarwar et al., 2012), motivated further investigation of IL-6 signaling as a therapeutic target for CAD (Ridker and Rane, 2021). Several heuristics may aid in linking an identified genetic variant to the gene it causally influences, including the proximity of the variant to the gene and the functional consequence of the variant (Forgetta et al., 2022). With the increasing availability of exome data, a 'gene-first' approach utilizes the gene as the unit of analysis (Backman et al., 2021). In these analyses, loss-of-function or deleterious missense variants are aggregated and tested for their associations with the outcome of interest. A causal link with the gene can more confidently be made in this setting. Multi-omics *in silico* data may also be integrated to help identify causal genes. One such approach prioritizes causal genes using machine learning methods to integrate multimodality functional data from proteomics, transcriptomics, and epigenomics across multiple tissue and cell types (https://genetics.opentargets.org/; Mountjoy et al., 2021). Additional methods for linking genetic variant to causal gene, and the respective strengths and weaknesses of these methods, are discussed at length elsewhere (Gallagher and Chen-Plotkin, 2018).

The drug target may also be identified from preclinical hypothesis-driven studies, or from high-throughput in vitro or in vivo screens. Results from hypothesis-free scans utilizing libraries of genetic proxies for protein levels can also be used to prioritize a drug target (Folkersen et al., 2020; Henry et al., 2022). Finally, an established drug target may be investigated for drug repurposing opportunities.

## Approaches to the identification of a genetic proxy for use in drug target MR analyses

The first step in designing a drug target MR analysis is to select a strategy for the identification of a genetic proxy for a drug target. This involves several decisions regarding location of the genetic variants in relation to the protein-coding gene sequence, the functional consequence of the variant, and the choice of phenotype used to weigh the effects of the genetic proxy. Genetic variants are *cis*-acting when they are located within or close to the gene of interest. This distance from variant to gene is not standardized, with some studies using variants

within 100 kb (Daghlas et al., 2021) upstream or downstream of the gene, and others using up to 1 Mb ranges (Pietzner et al., 2021; Yang et al., 2021). Efforts have been made to empirically define the optimal variant to gene distance for determination of *cis* versus *trans* function, but this is still a work in progress (Fauman and Hyde, 2022). *Trans*-acting variants are positioned outside this genomic range, are not as confidently linked to the gene of interest and are thus considered to be less robust proxies for use in drug target MR analyses (Swerdlow et al., 2016; Gill et al., 2021). This is in part due to the potential for the variant to influence genes and biological pathways independent of the drug target of interest and hence violate the exclusion restriction condition (see Said et al. (2022) for an example of this phenomenon with MR analyses of C-reactive protein [CRP]). As previously outlined, plausibility that the genetic proxy for the drug target relates to function of the gene of interest may be enhanced by selecting damaging or predicted loss-of-function variants in the protein-coding sequence of the gene (Deboever et al., 2018; Emdin et al., 2018; Daghlas et al., 2021). However, empiric studies have shown that reliable inference may still be obtained from MR analyses that only leverage intronic variants (Schmidt et al., 2020).

The second consideration is what phenotype to use to weigh the effect of the genetic variant (Figure 3). One possibility is to use a phenotype of molecular function, such as gene expression or protein abundance (where the variants are known as protein quantitative trait loci, or pQTLs; Porcu et al., 2019; Zheng et al., 2020). Of these, pQTLs measured in the relevant tissue may be preferable given the proximity of protein levels to clinical phenotypes (Schmidt et al., 2020; Gill et al., 2021), and the fact that drug targets are typically proteins. Several large genome-wide association studies (GWAS) have been published that catalog pQTL associations in different tissues and have made their results publicly available (Yao et al., 2018; Zheng et al., 2020; Pietzner et al., 2021; Suhre et al., 2021; Yang et al., 2021). A key limitation in the analytic application of pQTLs is confounding due to variant effects that interfere with the aptamer assay used to measure protein levels (Suhre et al., 2021). These biases may be examined by using, when available, multiple independent pQTL datasets that utilize different assays for measurement of protein levels (Zheng et al., 2020). Finally, molecular phenotypes may include measures of target engagement, such as the use of an acute phase reactant like CRP when examining effects of perturbing IL-6 signaling (Georgakis et al., 2021).

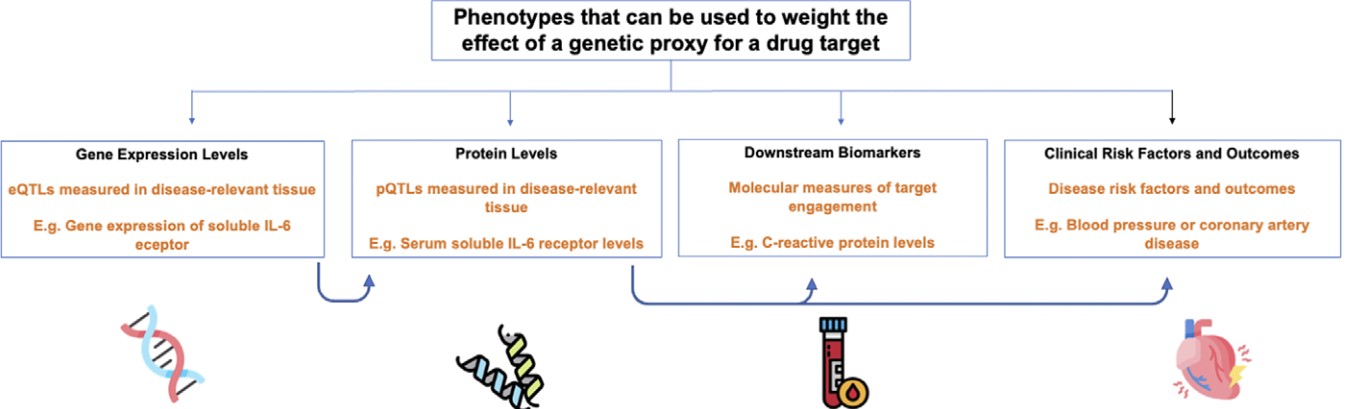

**Figure 3.** Different approaches to weighting the effect of a genetic variant on a drug target. The absence of an arrow between the 'molecular biomarkers' and 'clinical risk factors and outcomes' box demonstrates that a molecular biomarker does not necessarily have to be causal for a disease outcome.

An alternative or complementary strategy for weighting genetic variants for use in MR is to use disease risk factors or outcomes. An advantage of these approaches relative to molecular phenotypes is confirmation that the variant influences the clinical phenotype of interest. Examples of disease risk factors with relevance to drug targets include glycated hemoglobin for proxying effects of antidiabetic drugs (e.g., glucagon-like peptide 1 receptor agonists; Daghlas et al., 2021) and blood pressure for proxying effects of anti-hypertensive drugs (e.g., beta-blockers) (Gill et al., 2019). Alternatively, genetic variants may be weighted by their association with a binary disease outcome such as hypertension (Burgess and Labrecque, 2018). The genetic variants must be strongly and reliably associated with this phenotype of interest to meet the first MR assumption of relevance (Davies et al., 2018). In MR analyses, this statistical strength can be measured using the *F*-statistic (Davies et al., 2018). If multiple variants are used, they should either be independently inherited, or specialized methods should be employed to account for their correlation (Yavorska and Burgess, 2017). Such methods can, for example, use correlation matrices to account for linkage disequilibrium between genetic variants, thus ensuring that each genetic variant provides 'unique' information for the analysis.

Several analytic approaches have been used to proxy inhibition of IL-6 signaling. One approach is to use the rs2228145 variant, which is a missense variant that reduces membrane-bound IL-6 receptor levels, increases soluble IL-6 receptor levels, which serves as a decoy receptor that negatively regulates IL-6 signaling (Sarwar et al., 2012; Ferreira et al., 2013). A second approach is to use independent *cis* variants associated with CRP, an established inflammatory biomarker of IL-6 signaling (Georgakis et al., 2020). A third approach is to use pQTLs for soluble IL-6 receptor levels (Rosa et al., 2019). These approaches are not mutually exclusive and may be combined in complementary sensitivity analyses.

### Phase 1 and 2: Using MR to identify measures of target engagement

Once a genetic proxy for perturbation of a drug target has been identified, the association of the proxy with circulating molecules may be characterized to understand the biology of perturbation of the drug target and to identify target engagement biomarkers (Stefaniak and Huber, 2020). Such biomarkers may be considered for further investigation as measurable outcomes in clinical studies. This genetic approach has been applied to characterize the molecular signature of inhibition of IL-6 signaling. For instance, the IL-6 receptor antagonist tocilizumab is known to raise circulating IL-6 and soluble IL-6 receptor levels while reducing serum fibrinogen levels (Georgakis et al., 2020). An MR analysis using a *cis* genetic proxy for inhibition of IL-6 signaling recapitulated these same effects, validating use of this genetic proxy for testing the effects of perturbation of IL-6 signaling (Georgakis et al., 2020). In a more novel application of MR for this aim, a genetic proxy for IL-6 signaling was tested for its effects on 40 circulating cytokines to characterize the inflammatory signature of IL-6 signaling (Rahman et al., 2022). Significant associations of genetically proxied inhibition of IL-6 signaling were identified for ten inflammatory molecules, providing novel insights into the molecular signature of pharmacologic inhibition of IL-6 signaling.

### Phase 1 and2: Use of MR to assess for adverse effects of drug target perturbation

Genetic association data may be used in MR analyses for the investigation of on-target safety outcomes. On-target toxicity represents effects secondary to modulation of the drug target and can be investigated using MR. In contrast, off-target toxicity represents effects attributable to pleiotropic biochemical effects of the drug compound itself on proteins and pathways independent of the drug target of interest and cannot be investigated using MR (Rudmann, 2013).

One approach, known as a 'phenome-wide association study' (PheWAS), uses MR to perform hypothesis-free scans for adverse effects of drug targets across the human phenome (Denny et al., 2010). Outcomes in PheWAS are often defined using International Classification of Disease (ICD) codes and therefore sample sizes may be smaller than those available from large-scale GWAS consortia. In the case of genetically proxied inhibition of IL-6 signaling, a PheWAS identified safety signals for increased risk of atopic dermatitis, cellulitis, urinary tract infections, and cholecystitis. In such cases, it may be unclear whether adverse effects of a drug target outweigh its benefit. One approach to address this question is to investigate the outcome of lifespan (Daghlas and Gill, 2021). In the considered population, a net benefit in lifespan due to perturbation of a drug target suggests that adverse effects do not outweigh the clinical efficacy outcome and that the drug target may yield a mortality benefit in clinical trials (Daghlas and Gill, 2021). Using the outcome of parental lifespan, an MR analysis that demonstrated a possible lifespan benefit for genetically proxied inhibition of IL-6 signaling in a general population (Rosa et al., 2019). This finding supports the notion that adverse effects of inhibition of IL-6 signaling (e.g., infection) are outweighed by the benefits of inhibiting this pathway (e.g., reduced risk of cardiovascular disease). It is important to appreciate that these analyses are typically performed using genetic associations with lifespan in the general population (Timmers et al., 2019) and therefore may not accurately represent effects anticipated in population subgroups targeted in clinical trials.

### Pre-phase 3: Informing clinical trial design

#### Constructing a primary outcome

Investigators designing a clinical trial must select a primary efficacy outcome which is often a composite of related clinical outcomes (Andrade, 2015). MR may be leveraged to test the drug target across multiple related outcomes, and this information may be used to prioritize outcomes to include in a primary efficacy outcome in a clinical trial (Gill and Burgess, 2020). This is helpful because the inclusion of an outcome unaffected by the drug reduces statistical power and increases the odds of failure of a drug candidate in a phase 3 trial. For instance, the concordant effects of genetically proxied inhibition of IL-6 signaling on risk of stroke and CAD suggest that combining these clinical outcomes in a clinical trial could increase statistical power of a phase 3 clinical trial (Georgakis et al., 2020). In contrast, the null associations with venous thromboembolism suggests that inclusion of this phenotype in an efficacy outcome would reduce the statistical power of a clinical trial (Georgakis et al., 2020).

#### Investigating drug effects in population subgroups

Investigators and clinicians are typically interested in the effect of a drug in particular population subgroups. For instance, a clinical trial of an IL-6 inhibitor may enroll a patient population with evidence of

elevated levels of IL-6 driven inflammation. Similarly, clinical trials typically stratify their enrolled population across covariates of interest to test for consistency of treatment effect in patient subgroups. MR analyses may be designed to mimic these conditions and to test for heterogeneous effects across subgroups. This application is demonstrated by an MR analysis of genetically proxied inhibition of IL-6 signaling that stratified across numerous covariates including sex, age, kidney function, body mass index, LDL-cholesterol, blood pressure, and hemoglobin A1C (Georgakis et al., 2022). The relative benefit of inhibition of IL-6 signaling was consistent across strata of all covariates except for LDL-cholesterol, where a greater magnitude of benefit was seen at levels of LDL-cholesterol greater than 160 mg/dL (Georgakis et al., 2022). A larger absolute benefit was observed in the population subgroup with highest baseline high-sensitivity CRP levels, suggesting that clinical trials should prioritize this population for IL-6 inhibition therapies. MR may also be used to identify population subgroups that are less likely to benefit from perturbation of a given drug target. This is illustrated by the null effect of genetically proxied inhibition of IL-6 signaling on risk of the cardioembolic stroke subtype relative to the small and large vessel ischemic stroke subtypes (Georgakis et al., 2020).

### Testing for drug–drug interactions

New drugs are often used in concert with other drugs, such as the combination of antiplatelet therapies with statins for the secondary prevention of ischemic stroke. Indeed, polypharmacy is becoming more common despite the absence of evidence for the safety of drug combinations (Oktora et al., 2019). The potential consequences of such drug–drug interactions may be queried in appropriately designed MR analyses. One approach is to perform a $2 \times 2$ factorial analysis stratified across levels of genetic proxies for two drug targets (see the citation for alternative modeling approaches; Rees et al., 2020). This approach can determine whether genetically proxied drug effects on a clinical outcome are additive, or if they have supra-additive effects. An additive effect was identified in a $2 \times 2$ factorial analysis investigating interactions between genetically proxied inhibition of IL-6 signaling and genetically proxied *HMGCR* (the target of statins) inhibition (Georgakis et al., 2022).

### Phase 3: Use of genetic proxies in MR to test clinical efficacy of a drug target

Once a genetic proxy for a drug target of interest has been selected, statistical associations of the genetic variants with the outcome of interest may be extracted from a GWAS. The effect of drug target perturbation, weighted by changes in the levels of a relevant biomarker, may then be estimated using conventional statistical methods (Burgess et al., 2017; Hemani et al., 2018; Teumer, 2018). This approach was used in an MR analysis to support a protective effect of inhibition of IL-6 signaling on risk of the efficacy outcome of CAD (Sarwar et al., 2012). Subsequent studies identified similar effects for additional cardiovascular outcomes including abdominal aortic aneurysm, peripheral artery disease (Levin et al., 2021), atrial fibrillation (Rosa et al., 2019), and ischemic stroke (Georgakis et al., 2020). In contrast to the above studies, MR may be used to identify drug targets that are less likely to causally affect an outcome of interest. For example, MR analyses did not support causal effects of IL-6 signaling on pulmonary arterial hypertension, which was concordant with a null effect from a phase 2 clinical study (Toshner

et al., 2022). This result supports the redirection of resources away from further testing of IL-6 inhibition for this indication.

### Post-phase 3: Using MR to identify repurposing opportunities for an established drug

There is great interest in drug repurposing (also referred to as drug repositioning), whereby a drug that has undergone safety and efficacy testing for one indication is proven to be effective for a separate indication (Glenn Begley et al., 2021). Such an approach circumvents several challenges and expenses in drug development outlined in the Introduction (Pushpakom et al., 2018). MR may be used to provide evidence for these repurposing efforts (Gill and Vujkovic, 2022). This approach requires identification of a genetic proxy that is validated to influence the clinical outcomes for which a drug target has been approved (Gill and Burgess, 2020). A clinical outcome may then be selected for investigation, or a hypothesis-free PheWAS may be performed to identify novel repurposing opportunities.

As an example of a hypothesis-driven approach, genetically proxied inhibition of IL-6 signaling was investigated for its effects on the outcome of polymyalgia rheumatica (PMR), an inflammatory musculoskeletal disorder (Zhao and Gill, 2022). A protective effect of inhibition of IL-6 signaling was identified for this disease outcome, a finding which has been corroborated by clinical trial evidence for a beneficial effect of tocilizumab on disease activity in PMR (Devauchelle-Pensec et al., 2022). This approach was also used to identify a protective effect of genetically proxied IL-6 signaling on risk of COVID-19 incidence and severity, a finding consistent with results from clinical trials of IL-6 receptor inhibitors (Bovijn et al., 2020; Larsson et al., 2021; Rajasundaram et al., 2022). As an example of a hypothesis-free approach, an MR-PheWAS analysis using genetic data from the Million Veterans Program identified a potential repurposing opportunity for IL-6 receptor antagonists for prevention of aortic aneurysm (Cai et al., 2018), a finding replicated in the UK Biobank dataset (Georgakis et al., 2021).

### Limitations and context

Despite the potential for MR to inform drug development efforts, the methodology has several limitations that must be considered when contextualizing any findings. The first set of limitations relate to interpretation of the effect size from an MR analysis. First, these numeric estimates reflect the consequence of lifelong perturbation of a drug target. In effect, the 'time zero' for the natural experiment is set either at gametogenesis or when the variant becomes biologically relevant, and so the magnitude of effect over this duration may not be predictive of the magnitude of benefit from a clinical intervention of shorter duration. This is illustrated by the larger magnitude of effect estimates for LDL-cholesterol lowering from MR analyses relative to those from clinical trials (Ference et al., 2017). Second, MR estimates may be biased by canalization, whereby lifelong genetic effects on a phenotype are buffered by compensatory developmental processes and hence may differ from shorter-term targeting of a protein later in life (Lawlor et al., 2008). Third, effect estimates in conventional MR analyses correspond to differences in exposure levels around the population mean and cannot inform the consequences of large changes in levels or function of the drug target. Exceptions to this principle include variants with large magnitudes of effect, such as protein-truncating variants, or the use of analyses that employ nonlinear statistical methodologies (Burgess et al., 2014).

A second set of limitations correspond to the assumption of no bias due to pleiotropy, or the exclusion restriction condition. For instance, a genetic variant may be a *cis* pQTL for a drug target of interest and a *trans* pQTL for another protein. The association of this variant with a given outcome may be mediated through pathways independent of the protein of interest. Given the widespread pleiotropy in the human genome (Verbanck et al., 2018), the exclusion restriction assumption may be violated in many instances, although careful analysis may mitigate this bias. For instance, numerous statistical methods have been developed that recover consistent causal effect estimates despite varying degrees of pleiotropic effects of the genetic variants (Bowden et al., 2017; Hemani et al., 2018). Additionally, *cis* variants are generally less likely to be pleiotropic than *trans* variants (Schmidt et al., 2020). Finally, databases (Kamat et al., 2019) of genetic associations may be used to test for pleiotropic associations of the genetic variants used as proxies.

In some cases, there may not be genetic variants available to proxy a drug target of interest. This can be due to a lack of sufficient human genetic variation in the gene region that influences the phenotype of interest within a given ancestry group, or there may not be variants at a locus that meet the assumptions for a valid MR analysis (e.g., at a highly pleiotropic locus). This limits the number of genes available for potential investigation using MR. Further, the selection of genes to be investigated is typically guided by the hypothesis or scientific question being addressed in the analysis. Additional limitations include lack of availability of a GWAS of the relevant phenotype or biomarker for that drug target, or lack of a GWAS of molecular data from the appropriate tissue or cell type. Another related limitation is that MR cannot necessarily instrument a drug that concurrently influences multiple parallel biological pathways. Rather, molecular mediators may be proxied individually, or in combination using a factorial MR approach (Yarmolinsky et al., 2020).

Additionally, target phenotypes, disease outcomes, or disease subtypes may not yet be available as outcomes in sufficiently large genetic association datasets. For certain outcomes such as heart failure, clinical trials test interventions separately in patient subgroups, such as in patients with preserved, rather than reduced ejection fraction (Anker et al., 2021). In such cases, MR cannot yet be implemented to test hypotheses that parallel those investigated in clinical trials. Finally, in contrast to studies of disease susceptibility, data for genetic predictors of disease progression or disease outcomes are scarce (Paternoster et al., 2017).

The conventional principles that guide interpretation of observational research also apply to MR analyses. Statistical power and precision of effect estimates should always be considered when interpreting a null result, particularly for outcomes with small sample sizes. Most genetic studies are performed using data from individuals of European ancestry and are therefore of unclear generalizability to individuals of different ancestries. This limitation may be addressed by the inclusion of diverse ancestry groups in future GWAS of clinical and molecular traits. When possible, independent replication and triangulation (Lawlor et al., 2016) of results with those from orthogonal research methodologies enhances confidence in any given finding. Finally, the strength of inference from an MR analysis is heavily influenced by the quality of the phenotype used in genetic association analyses. Modern statistical techniques and large sample sizes cannot overcome biases created by misdiagnosis, diagnostic or phenotypic heterogeneity, and ascertainment bias.

At this point, it is worth noting the similarities and differences between an MR finding and a genetic association at the same locus. For example, GWAS of CAD identify genetic variants in *IL6R*. (Deloukas et al., 2013). Indeed, such an association could be used

as evidence for a causal effect of IL-6 signaling on CAD risk. The MR paradigm offers several additional benefits. First, the MR paradigm explicitly formalizes the assumptions for causality and provides methods to test the plausibility of the assumptions. For example, numerous sensitivity analyses have been developed that provide results that are robust to inclusion of variants that affect the outcome through pathways unrelated to the exposure (Bowden et al., 2015, 2016). Second, MR can be used to produce effect estimates weighted by changes in levels of a clinical phenotype or biomarker. Third, MR can be used to aggregate the effects of multiple independent genetic variants and hence identify effects that are not genome-wide significant when using a single variant in a GWAS. Fourth, in hypothesis-driven MR the association of the variants with the exposure typically does not require as stringent of a statistical significance threshold as does a GWAS for a given locus (Davies et al., 2018). Thus, findings from drug target MR analyses should be viewed as complementary to results from a genetic association at a given locus.

## Conclusion

There is great potential for MR to aid in the drug development process. Ongoing clinical trials of inhibition of IL-6 signaling and for other drug targets prioritized in MR analyses will ultimately serve as tests for the utility of MR in drug development efforts and the advancement of human health.

**Open peer review.** To view the open peer review materials for this article, please visit http://doi.org/10.1017/pcm.2023.5.

**Data availability statement.** Data analysis was not performed for this study.

**Acknowledgements.** The figures were designed using open-access images from https://www.flaticon.com/.

**Author contributions.** I.D. and D.G. both made substantial contributions to the conception or design of the work; or the acquisition, analysis, or interpretation of data for the work. I.D. and D.G. both contributed to drafting the work or revising it critically for important intellectual content. I.D. and D.G. both provided final approval of the version to be published. I.D. and D.G. both agree to be accountable for all aspects of the work in ensuring that questions related to the accuracy or integrity of any part of the work are appropriately investigated and resolved.

**Financial support.** D.G. is supported by the British Heart Foundation Centre of Research Excellence (RE/18/4/34215) at Imperial College London.

**Competing interest.** I.D. declares no conflicts of interest. D.G. is employed part-time by Novo Nordisk.

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
