## [Reviewer Report]

*Comments to Author*: This article gives and overview how Mendelian Randomization (MR) can be utilized to inform the different stages of drug development. Throughout the review the authors illustrate how MR applications using genetic inhibition of interleukin-6 as an exemplar.

1. The main MR examples in the manuscript use interleukin-6 (IL6) but in the preamble PCSK9 is used as an example – what was the rationale for this? Is the information on PCSK9 needed?

2. How were the published genetic studies for IL-6 that are included in Table 1 selected? Was this a done systematically?

3. On line 185 the authors state that “Additional methods for linking genetic variant to causal gene, and the respective strengths and weaknesses of these methods, have been discussed at length elsewhere”. It would be helpful to include a brief description of the methods or at least summarize the approaches in a table.

4. On lines 248- 250 the authors state that “If multiple variants are used, they should be independently inherited, or methods designed for the use of correlated genetic variants should be employed.” Although a reference is given it would be helpful to provide a brief summary of the methods should be provided as the target audience for this article will be non-experts.

5. More details on how lifespan can be used to demonstrate net benefits is needed. An example of the findings in relation to IL6 should be given rather than just citing a reference with minimal detail.

6. I would have liked to see included in the review with recommendations on how the representation of different ancestries can be improved – not just that these studies are lacking and so generalizability is limited to Europeans only.

---

## [Reviewer Report]

*Comments to Author*: The review concisely explains the concept of Mendelian randomization (MR) and places it into a drug discovery context with a series of exemplars focusing on IL6 signalling.

The review lays out some of the decision making process during the design of MR experiments, particularly in the selection of the genetic instrument. For example the variants should generally by cis to the gene, and ideally in the coding region so that the causal link is strongly established.

The review considers in detail some of the key elements that are needed to plan a successful MR study, including a clearly defined modifiable exposure of interest, the outcome of interest, the genetic instrument, and the statistical analysis required to detect a true signal.

Overall this is an excellent review and I agree with the authors that growing international population genetic resources will certainly increase the use of MR so that it becomes a pivotal tool in most drug development efforts.

Minor comments that might be addressed by the authors:

1) One area I would like to have seen addressed a little more thoroughly, was a comparison of MR with conventional genetic association methodology. I understand that MR is considered a more powerful method for studying the causal effect of an exposure on an outcome because it can provide a more unbiased estimate of the causal effect and is less prone to confounding compared to observational studies, but it would be good to contrast this to more general use of genetic association. In short how do an MR finding and a genetic association at the same locus differ?

2) Another area for discussion is the suitability of a locus for MR? Is it likely that all genes could potentially be investigated using MR? What are the likely key limitations and would it be possible to estimate roughly how many genes might be investigated? Which disease traits are particularly suited to MR and conversely are there disease traits that are unsuited to a MR approach?

---

## [Editor Report]

*Comments to Author*: As you will see from their comments, the reviewers were positive about your review but raise a few minor questions and suggestions that will hopefully improve the final paper when addressed.

---

## [Reviewer Report]

*Comments to Author*: This is an excellent review that serves as an informative introduction to the use of MR. I will not reiterate my previous comments, but note that my comments about the need for comparison and contrast between MR and GWAS findings are fully addressed.